# Mental Health of PhD Students at Polish Universities—Before the COVID-19 Outbreak

**DOI:** 10.3390/ijerph182212068

**Published:** 2021-11-17

**Authors:** Mateusz Kowalczyk, Michał Seweryn Karbownik, Edward Kowalczyk, Monika Sienkiewicz, Monika Talarowska

**Affiliations:** 1Babinski Memorial Hospital, Aleksandrowska St. 159, 91-229 Lodz, Poland; mateuszjerzykowalczyk@gmail.com; 2Department of Pharmacology and Toxicology, Medical University of Lodz, Żeligowskiego St. 7/9, 90-752 Lodz, Poland; michal.karbownik@umed.lodz.pl (M.S.K.); edward.kowalczyk@umed.lodz.pl (E.K.); 3Department of Pharmaceutical Microbiology and Microbiological Diagnostic, Medical University of Lodz, Muszyńskiego St. 1, 90-151 Lodz, Poland; 4Department of Clinical Psychology and Psychopathology, Institute of Psychology, University of Lodz, Smugowa St. 10/12, 91-433 Lodz, Poland; monika.talarowska@now.uni.lodz.pl

**Keywords:** mental health, PhD students, depression, anxiety, insomnia

## Abstract

Background: A group particularly exposed to the occurrence of disorders in the sphere of the psyche are young people with a newly developing personality structure and a sense of identity. In the available literature there are few reports describing the mental health of doctoral students—a group that is affected by a particular group of stressors. The aim of the research was to assess the mental health of PhD students at Polish universities. Material and Methods: The Polish adaptation of the GHQ Questionnaire-28, developed by David Goldberg et al. was used in the research. PhD students from all universities associated in the National Representation of Doctoral Students were invited to take part in the research. A total of 576 completed questionnaires were received. Results: It was found that depression is statistically more frequent in doctoral students who are not in any relationship with another person; anxiety/insomnia is more common in women than men and less frequently in doctoral students of general than in technical universities. Conclusions: (1) More than half of the surveyed students complain about the deterioration of mental health. The most commonly reported symptom groups are anxiety and insomnia, followed by social dysfunctions and somatic symptoms. (2) Depression is statistically more common in people who are not in any relationship with another person and anxiety and insomnia are statistically more common in women than in men and statistically less frequent in doctoral students of general universities than technical universities. (3) In view of the presented results, educating young adults in the field of self-awareness in the field of mental health seems to be particularly important.

## 1. Introduction

In recent years, a real epidemic of mental disorders has been observed in the world, which worsens the quality of lives of those affected, their families and beloved ones. A similar situation can also be observed in Poland [1,2,3]. We live in a state of permanent escalation of expectations, and we are overwhelmed by tasks that never end. This makes us perceive our life as being difficult [4].

A group particularly exposed to the occurrence of disorders in the sphere of the psyche are young people with a newly developing personality structure and a sense of identity [5,6]. According to Marcon et al. [7], 53.03% of medical students (N = 4840) admitted to high-risk drinking. According to Unwin et al. [8] and Fond et al. [9], other common mental disorders in young adult learners include depressive, anxiety, sleep, and eating disorders.

Stecz and Podgórska-Jachnik in the work “Mental health (also students): commentary to the EZOP Poland” [10] reported that when it comes to Polish students, depressive episodes occur in about 1.3% of cases, and specific phobias in 5.8%. Jasiński [11], while investigating the mental health of physical education students, noticed that they statistically significantly more often revealed the symptoms of depression than the somatically healthy people. Equally often are the symptoms of burnout [12]. Moreover, a study by Morris et al. [13] proved that over the last decade from 2007 to 2018–2019, there was an increase in the use of nearly all classes of psychiatric medications: antidepressant medication use increasing from 8.0% to 15.3%, anti-anxiety medication from 3.0% to 7.6%, psychostimulants from 2.1% to 6.3%, antipsychotics from 0.38% to 0.92%, and mood stabilizers from 0.8% to 2.0%.

Young people worry about their job: they cannot find it or if they have it, they are afraid to lose it. Therefore, they often take on additional responsibilities to prove themselves, but in the end, they are unable to meet them. The necessity of the continuous professional development which employers impose on employees means that people forget about the necessary rest. The consequences of such a lifestyle are more frequent symptoms from the mental sphere [14,15].

The aim of the present study is to assess the mental health status of PhD students of Polish universities (i) and check whether particular types of university students differ in this respect between one another (ii). We chose this group because we believe that this group has special requirements resulting from the necessity to combine scientific work with practice.

## 2. Material and Methods

### 2.1. Material

A total of 830 questionnaires were distributed, 576 of which were returned, 528 were found complete, which accounted for 63.6% of the response rate. At the stage of recruiting study participants, a purposeful selection was used (PhD students from all universities associated with the National Representation of Doctoral Students were invited to take part in the research). PhD students from technical universities, medical colleges, and universities participated in the study. The subjects were qualified to participate in the study after giving their written informed consent. The sample size was estimated a priori on the basis of G * Power 3.1. [16], based on the predicted types of statistical analyzes, the number of subgroups, and variables. The characteristics of the tested sample are presented in Table 1.

### 2.2. Methods

#### 2.2.1. Procedure of the Study

During the meeting of the National Representatives of Doctoral Students, 830 General Health Questionnaires (GHQ) developed by David Goldberg and associates [16] were distributed in the GHQ-28 version [17] among the Chairpeople of the Doctoral Students’ Associations of Polish universities (which expressed their willingness to participate in the research). The number of sheets downloaded depended on the Chairpeople of Local Governments—there were 1000 questionnaires. At home universities, volunteers were given a questionnaire to be filled in on their own.

#### 2.2.2. Psychological Examination

The GHQ-28 questionnaire is used to assess the mental health of adults. The intention of its authors was not to make a thorough psychiatric diagnosis but to determine the probability, occurrence, and the possibility of developing a mental disorder. 

The GHQ-28 questionnaire allows clinicians to select people from a given population who are at risk of developing mental health disorders. The GHQ-28 version has four scales, each of which includes seven items: A: somatic symptoms, B: anxiety, insomnia, C: social dysfunction, D: severe depression. The GHQ-28 questionnaire belongs to the so-called self-reports in which each item is accompanied by four possible responses (not at all, no more than usual, rather more than usual, much more than usual). In the questionnaire, the categories of answers appear in a column layout and are assessed as follows: 0-0-1-1. Thresholds are set for the result obtained by this method (for GHQ-28, they are set at 5/6), which are the basis for identifying people with disorders among all patients. The overall score in GHQ-28 is the sum of the points obtained for the replies to all questions in the questionnaire and can be as high as 28.

#### 2.2.3. Statistical Analysis

The dependent variables presented non-normal distribution and were treated as ordinal. Thus, they were presented as median, 25th, and 75th percentile, if not stated otherwise. Categorical independent variables of more than two levels were converted to dummy variables. Missing data were excluded and the analyses were based on complete cases only. The Mann–Whitney *U* test was used to test the differences in GHQ-28 subscales between independent categorical variables. The Spearman’s rank correlation coefficient was used for that purpose, for at least ordinal independent variables. The false discovery rate was controlled at the level of 0.05 with the Benjamini and Hochberg correction for testing multiple hypotheses. Pearson’s chi-squared test and the two-way analysis of variance (ANOVA) were further used for exploratory purposes. *p*-values below 0.05 were considered statistically significant. The analysis was performed using STATISTICA 13.1 Software (StatSoft, Tulsa, OK, USA).

#### 2.2.4. Bioethics

The approval of the Bioethical Commission of the Medical University of Lodz, number RNN/129/18/KE of 10 April 2018 for conducting the research has been obtained and participants have given consent for their data to be used in the research.

## 3. Results

### 3.1. The Results of the Study with Use of GHQ Scales

#### 3.1.1. General Analysis

The number and the percentage of the study participants showing five or more points in each of the GHQ scales are presented in Table 2. All 576 participants were included in the analysis (also those with not completed data sheets).

After an adjustment for testing multiple hypotheses, three statistically significant associations were found in the GHQ-B subscale and one in the GHQ-D subscale. The anxiety/insomnia parameter measured with the GHQ-B subscale was higher in female students than males (2(0–4) vs. 1(0–3), *p* = 0.0094), lower in students of universities than students of other schools (1(0–3) vs. 2(0–4), *p* = 0.0081), and lower in students of human science than others (0(0–3) vs. 2(0–4), *p* = 0.0043). The Benjamini–Hochberg corrected significance level for testing the associations of the variables with the GHQ-B subscale was 0.0094.

As studying human sciences is typical at general universities, the respective variables were found collinear (χ2(1) = 80.6, *p* < 0.0001), and the latter two associations may be self-explainable. On the other hand, sex of the students is probably not associated with studying human sciences (χ2(1) = 0.2, *p* = 0.63), and a further exploration suggests they both independently contribute to associating with anxiety/insomnia (two-way ANOVA yielded both sex and studying human sciences as significantly associated with the dependent variable (F(1.347) = 5.24, *p* = 0.023, and F(1.347) = 5.09, *p* = 0.025, respectively) and with a non-significant effect of their interaction (F(1.347) = 0.07, *p* = 0.80). However, the full model explained only 3.1% of the variance in the measured anxiety/insomnia parameter, suggesting a minor effect of the tested independent variables. Severe depression measured with the GHQ-D subscale was lower in students involved in any partnership (informal or formal) than in single students (0(0–0) vs. 0(0–1), *p* = 0.0051). The Benjamini–Hochberg corrected significance level for testing the associations of the variables with the GHQ-D subscale was 0.0094.

#### 3.1.2. Analysis of the Subscales of the GHQ-28 Test

In addition, analyzing 576 GHQ-28 health questionnaires according to the responses provided, it was noted that somatic symptoms (A) were a problem for about 15% of the respondents, “anxiety and insomnia” (B) for about 20%, functional disorders (C)—for around 15%, symptoms of depression (D)—for about 5% of the doctoral students. The highest number of points in part A of the questionnaire—about 48% of the respondents —was obtained from those who felt exhausted and weakened more or much more and 46% from those who felt the need to improve their state more or much more. In part B, in turn, 46% of the students indicated that they felt overwhelmed by the excess of duties and 41% reported they were constantly overworked. In part C, however, it was most often (34%) indicated that students can less or much less than usual enjoy their ordinary everyday life and are less or much less satisfied with the performance of their tasks (30%). In part D of the questionnaire, 14% of the respondents indicated that they considered themselves worthless more than usual and much more than usual and that they could not do anything about it due to the bad condition of their nerves. Moreover, 7% of the respondents would rather more or much more opt for ending their own lives and 6% claimed that the thought of committing suicide kept crossing their minds.

## 4. Discussion

The World Health Organization (WHO) [18,19] defines mental health as a state of well-being in which each person realizes their own potential, is able to cope with normal life stress, work efficiently and fruitfully, as well as contribute to the community. Mental health problems are a serious public health dilemma in many countries. About one-third of the disease burden in the United States is a mental disorder that most often occurs in people aged around 20 [18]. Yusof and Azman [20] conducted a study to find out the source of stress among students in institutions of higher education in Malaysia. Researchers found that students were stressed because of academic, financial, and time management problems. Stress reaches a critical level when students fail to solve these issues. Sense of belonging reduced doctoral students’ odds of clinically significant anxiety and depression symptoms, while academic stressors, relationships stressors, and financial stressors increased such odds [15]. According to Poh et al. [21], approximately 25% to 35% of students in Malaysia suffer from stress due to the increasing amount of work related to learning. Moreover, students are trying to meet parents’ expectations, which places high pressure on them [22].

In our study anxiety/insomnia parameter measured with the GHQ-B subscale (anxiety, insomnia) was higher in female students than males, lower in students of universities than students of other schools, and lower in students of human science than others. In the context of the sex of the respondents, the obtained results are consistent with the reports of other authors [23,24]. Women significantly more often complain of the experienced anxiety symptoms than men, which is probably related to their greater freedom in seeking professional medical help [25]. In the case of the surveyed men, the fear of severe stigmatization or shame [26]. For this reason, students can often hide their problems from those who are able to help them [20,27]. Perhaps because of this, in our research, we did not receive back more than 50% of the surveys (despite our requests), because some of the students may have decided that they prefer not to know the state of their mental health.

An interesting issue is the differences in the level of anxiety symptoms experienced by representatives of different types of schools and fields of science (lower in students of universities than students of other schools and lower in students of human science than others). There is a lack of research analyzing this issue, but it can be assumed that learning in the above-mentioned types of schools is associated with the lowest degree of emotional burden (lower faculty requirements, less workload, lower social pressure, and fear of failure if you fail) [28,29,30,31].

In the presented study, remaining in a romantic relationship turned out to be a protective factor against the onset of depression symptoms (severe depression measured with the GHQ-D subscale was lower in students involved in any partnership (informal or formal) than in single students). This result is consistent with the reports of other authors [32,33] and the views emphasizing the role of the lack of social support in the etiology of depressive disorders [29,34]. 

A report by the World Health Organization (WHO) [35] mentioned that the number of people with mental health problems will increase, and young people are among the highest risk group that can experience this problem. The problem of mental health among students has been analyzed by researchers all over the world [36]. Liu and colleagues [37] observed an increase in depression among American college students. The high exposure to various kinds of stress in the student population of the United States and the high impact of stress on mental health and suicides indicate the urgent need to establish a health care strategy for selected groups of students to improve their well-being. Gengoux and colleagues [38] note that even in medical schools there is a critical need for systematic advice on how to personalize prophylaxis and treatment programming to help students who are most at risk of mental disorders.

In our own research, the high (over 20 points) value is noted in 5% of the doctoral students, especially considering points from scales B (anxiety, insomnia), C (social dysfunction), and D (severe depression). According to the authors of the questionnaire [17], this may indicate that the students suffer from depressive mental health disorders with the concomitant risk of committing a suicide attempt. According to the responses, 7% of them would rather experience or even desire their own death and 6% have thoughts of taking their own lives.

The problem of suicides in the Polish population of adolescents and young adults was pointed out by Gmitrowicz and Dubla [39]. As a motive for a suicide attempt, 47.6% of respondents mentioned a lack of meaning in life, 30% trouble at school, 24.8% difficult family situation, 22.8% loss of a loved person, 19% fascination with death. According to the National College Health Assessment Survey [40], over 9% of students are considering suicide, but 80% of students who thought about suicide did not receive any mental health support at all.

In our study, the results obtained in the depression scale turned out to be particularly important (scale D of GHQ-28). Shamsuddin [41] discovered that depression is most experienced by students aged 20 to 24. Older students (aged 20–24) have higher depression rates compared to younger students (aged 18–19). A student who has problems with depression for various reasons can have lower academic performance. In other words, student academic achievement is threatened by depression [41].

In our research, we noticed that depression is statistically more common in students who are not in any relationship with another person. Anxiety and insomnia, in turn, were statistically more frequent in women than in men and statistically less frequent in doctoral students of general than technical universities. Many studies have shown that an anxiety disorder influences student performance at schools, colleges, and universities. For example, Shamsuddin et al. [41] have proved that there is a significant negative correlation between anxiety and achievement of grades obtained by students. These findings are comparable to what Sieber, O’Neil, and Tobias [42] discovered, that low academic achievement is attributed to high levels of anxiety which weakens the cognitive functions of students.

European [43,44,45] recommendations indicate the necessity to introduce model forms of mental health promotion at subsequent levels of education, including secondary and higher education. An example of good practice is a campaign for mental health implemented in Poland at the University of Silesia [46]. As a part of the campaign, discussions, film screenings, psychological workshops, psychological consultations, and art workshops are organized. Another example of activities in the area of mental health promotion addressed to students in Poland is the program implemented in 2010 at the Jagiellonian University. As a part of it, many debates and workshops have taken place. An educational platform has been also created, it contains materials and articles on mental health, addresses of centers providing support, and a list of certified psychotherapists [47]. The promotion of mental health, which aims to strengthen mental health and develop specific properties and skills of an individual, can be a chance for the development and satisfaction of young people [48]. A sense of competencies, strengths, and high self-esteem are examples of factors that strengthen the mental health of young adults and, thus, allow them to implement further development tasks, such as continuing education, starting work, and building long-lasting and satisfying relationships.

## 5. Limitations and Further Research Directions

The authors are aware of the influence of personality and temperamental traits on mental health, but the assessment of personality traits was not the aim of this study. Additionally, coping strategies and support mechanisms can also be important.

It should also be remembered that the presented results relate to the respondents living in specific socio-cultural conditions (Central Europe).

The study, which we present, was conducted in 2019 before the COVID-19 outbreak. Further research is needed to assess the impact of the pandemic on the mental health of PhD students [49].

## 6. Conclusions and Recommendations

More than half of the surveyed students complain about the deterioration of their mental health. The most commonly reported symptom groups are anxiety and insomnia, followed by social dysfunctions and somatic symptoms.Depression is statistically more common in people who are not in any relationship with another person and anxiety and insomnia are statistically more common in women than in men and statistically less frequent in doctoral students of general universities than technical universities.In view of the presented results, educating young adults in the field of self-awareness in the field of mental health seems to be particularly important.

## Figures and Tables

**Table 1 ijerph-18-12068-t001:** The characteristics of the tested sample.

Variable	Median (25th–75th Percentile) or n (Frequency)
**Sex**	
Female	293 (55.6%)
Male	235 (44.4%)
**Age (years)**	
28 (26–30)	
**Place of residence**	
Up to 50,000 inhabitants	63 (11.9%)
50,000 to 100,000 inhabitants	33 (6.3%)
100,000 to 250,000 inhabitants	48 (9.1%)
Over 250,000 inhabitants	3–3 (57.4%)
Village	81 (15.3%)
**Marital status**	
Single Partnership	243 (46.0%)285 (54.0%)
**Offspring**	
Yes	90 (17.0%)
No	438 (83.0%)
**Year of study**	
First	118 (22.2%)
Second	123 (23.4%)
Third	169 (32.2%)
Fourth and higher	118 (22.2%)
**Type of school**	
University	303 (58.2%)
Medical university	39 (7.4%)
Technical university	178 (33.8%)
Other	3 (0.6%)
**Fields of Science**	
Human science	145 (28.1%)
Social science	114 (21.3%)
Medical science	42 (7.7%)
Life science	45 (8.5%)
Technical science	141 (26.7%)
Physical science	33 (6.2%)
Agriculture and veterinary science	5 (0.9%)
Other	3 (0.6%)
**Professional Activity**	
Yes	363 (68.8%)
No	165 (31.2%)

**Table 2 ijerph-18-12068-t002:** Number and percentage of the study participants showing five or more points in each of the GHQ scales.

**A—somatic symptoms**	84 (14.6%)
**B—anxiety, insomnia**	114 (19.8%)
**C—social dysfunction**	89 (15.4%)
**D—severe depression**	30 (5.2%)
**A + B + C + D**	**291 (50.5%)**

## Data Availability

The data supporting reported results are available from the first author of article.

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
