# Peer review of "Mental Health of PhD Students at Polish Universities—Before the COVID-19 Outbreak"

_ijerph, 2021, doi:10.3390/ijerph182212068_

Round 1
Reviewer 1 Report
General Impression:
The subject of the work is very important, topical. Manuscript: "Mental health of PhD students at Polish universities - before the outbreak of the epidemic of COVID-19" requires significant changes, especially in the Introduction and Discussion section.
Detailed comments:
Introduction
According to the reviewer, the Introduction section should be redrafted and supplemented to relate to the purpose of the work.
Please explain why the authors chose such a group for research.
Line 12: "Equally often are the symptoms of burnout" - does this apply to physical education students? Please check.
Materials and methods
Reformat table 1 to make it more readable. Especially the row "Branches of science".
Results
Line 127: Invalid table number. Table 1 is on the line 84.
Line 125: Table II. - Invalid table format.
Lines: 147-164. Where are the results in this passage shown?
Discussion
The discussion relates to a small extent to the authors' results. It consists of fragments that do not form a common whole, so this part requires thorough editing.
Line 170. Add references.
Lines: 204-205. Check sentences / punctuation. Same on lines 208-209.
Lines: 225-228. Fragment needs some explanation.
Author Response
Thank you for your detailed review of our article. Regarding particular comments:
Ad 1. Introduction
According to the reviewer, the Introduction section should be redrafted and supplemented to relate to the purpose of the work.
Please explain why the authors chose such a group for research.
Line 12: "Equally often are the symptoms of burnout" - does this apply to physical education students? Please check.
We followed all the comments of the Reviewer (the changes are marked in the text). The aforementioned citation relates to medical students.
Ad 2. Materials and methods
Reformat table 1 to make it more readable. Especially the row "Branches of science".
Results
Line 127: Invalid table number. Table 1 is on the line 84.
Line 125: Table II. - Invalid table format.
Lines: 147-164. Where are the results in this passage shown?
We followed all the comments of the Reviewer (the changes are marked in the text).
Discussion
The discussion relates to a small extent to the authors' results. It consists of fragments that do not form a common whole, so this part requires thorough editing.
Line 170. Add references.
Lines: 204-205. Check sentences / punctuation. Same on lines 208-209.
Lines: 225-228. Fragment needs some explanation.
We followed all the comments of the Reviewer. The discussion was shortened and redrafted.
Sincerely,
MS
Reviewer 2 Report
Even before COVID, the mental health of postgrad students is a worthwhile study. The study reported is simply and well executed, however to increase usefulness of the study, I recommend exploring in greater depth the recommendations and presenting these recommendations in summary form in the conclusion to the study. Consider also in more detail what other limitations are present in the study - e.g. a particular sample of population in a particular country.Be wary of presenting conclusions in a way that suggests that the results are applicable to all post grad students, everywhere. Consider more specific attention to the role of supervisor and support mechanisms that might be put in place.
To enhance the article bring a greater practical perspective. The article holds potential to be very informative to those supervising post graduate students so frame the conclusion and discussion with this aim in mind.
On a minor point, please review the paragraphing in the article - there are some very long sections that need to be divided into paragraphs to improve readability and also to help weave a more cohesive 'story'.
Some interesting points were raised and it is good to see research in the area.
Author Response
Thank you for your detailed review of our article. Regarding particular comments:
Ad. 1. Even before COVID, the mental health of postgrad students is a worthwhile study. The study reported is simply and well executed, however to increase usefulness of the study, I recommend exploring in greater depth the recommendations and presenting these recommendations in summary form in the conclusion to the study.
We followed all the comments of the Reviewer (the changes are marked in the text).
Ad 2. Consider also in more detail what other limitations are present in the study - e.g. a particular sample of population in a particular country. Be wary of presenting conclusions in a way that suggests that the results are applicable to all post grad students, everywhere.
We followed all the comments of the Reviewer (the changes are marked in the text).
Ad 3. On a minor point, please review the paragraphing in the article - there are some very long sections that need to be divided into paragraphs to improve readability and also to help weave a more cohesive 'story'.
We followed all the comments of the Reviewer (the changes are marked in the text).
Sincerely,
MS
Round 2
Reviewer 1 Report
The work contains a huge number of errors, e.g. lines 43-48.
For example, the authors do not use the plural; "According" should be "According to" Do not use Polish words (line 46)
Such errors in the work suggest that the work has not been checked carefully and requires further correction.
According to the reviewer, the changes introduced in the discussion are not sufficient. The reviewer suggests that you conduct a thorough discussion of the results
Author Response
Dear Reviewer,
Thank you for your time and preparation of the review of our article.
We checked the work again and expanded the discussion as suggested.
Sincerely,
MS
This manuscript is a resubmission of an earlier submission. The following is a list of the peer review reports and author responses from that submission.
Round 1
Reviewer 1 Report
Thank you for giving me the opportunity to review this manuscript. This manuscript, entitled "Mental health of PhD students at Polish universities", aimed to assess the mental health of doctoral students at Polish universities.
Abstract: Background needs to be reformulated based on the study context. It is also important to highlight the type of study in methods. Rephrase the conclusion based on the results found.
Introduction: Does not highlight the research problem and purpose of the study. Little was said about what is known about the mental health of doctoral students.
Method: It remains to describe the type of studies, inclusion and exclusion criteria for the sample. Was a sample calculation performed?
Results: Weak.
Conclusion. The conclusions do not reflect the results presented.
The article in its current state is not suitable for publication.
Author Response
Dear Reviewer,
Thank you for your time and preparation of the review of our article. The introduced changes are marked in the text. In response to specific comments:
Ad. 1. Abstract: Background needs to be reformulated based on the study context. It is also important to highlight the type of study in methods. Rephrase the conclusion based on the results found.
The abstract was corrected according Reviewer comments.
Ad 2. Introduction: Does not highlight the research problem and purpose of the study. Little was said about what is known about the mental health of doctoral students.
The introduction was corrected according Reviewer comments.
Ad 3. Method: It remains to describe the type of studies, inclusion and exclusion criteria for the sample. Was a sample calculation performed?
The description of the study was reformulated. This section has been supplemented with missing information.
Ad. 4. Results: Weak.
The method of presenting the obtained results was reformulated (section Results).
Ad. 5. Conclusion. The conclusions do not reflect the results presented.
The conclusions were reformulated according Reviewer's comments.
Sincerely,
MS
Reviewer 2 Report
In this manuscript, the Authors investigated the prevalence of mental health issues in Polish Doctoral students. Furthermore, they describe associated factors and discuss implication of their results.
The manuscript is interesting and fills a gap in the literature (this is the first study investigating mental health in Polish PhD students).
I have few suggestions:
- The Introduction should be completely revised: they should focus on mental student’s mental health clearly stating the gap in the literature. Then they should indicate they research questions. As it is the Intro is very confusing, with repetitions. It should focus on education.
- Some of the Results are in the Discussion section
- The Discussion should focus on the results. The Author should describe the organization of the Polish PhD curriculum (duration, exams, dissertation…) and discuss differences and similarities with PhD programs in other EU Countries, US, and UK .
- There are important issues and relevant literature that were non considered in the Discussion. For instance, academic stress has been linked to abnormal brain plasticity in graduate students. The date is relevant because plasticity plays a pivotal role in the development of mental disorders. Please include and discuss the following reference:
1) Concerto C. Academic stress disrupts cortical plasticity in graduate students. Stress, Volume 20, Issue 2, Pages 212 – 216. 2017”
- Health risk behaviors often reported in graduate students (use of cannabis, energy drinks, stimulants) have been linked to mental distress in graduate students. The author should include the following reference and a statement regarding the need to include other relevant variables in future studies:
- Al Sawah M. Perceived stress and coffee and energy drink consumption predict poor sleep quality in podiatric medical students: A cross-sectional study. Journal of the American Podiatric Medical Association, Volume 105, Issue 5, Pages 429 – 434, September 2015
- One important aspect that was not considered are the temperament and personality traits which have been linked to mental distress in students across programs. They should include the following reference and include a statement in the Discussion:
- Conclusions: I would remove the bullet points
In this manuscript, the Authors investigated the prevalence of mental health issues in Polish Doctoral students. Furthermore, they describe associated factors and discuss implication of their results.
The manuscript is interesting and fills a gap in the literature (this is the first study investigating mental health in Polish PhD students).
I have few suggestions:
- The Introduction should be completely revised: they should focus on mental student’s mental health clearly stating the gap in the literature. Then they should indicate they research questions. As it is the Intro is very confusing, with repetitions. It should focus on education.
- Some of the Results are in the Discussion section
- The Discussion should focus on the results. The Author should describe the organization of the Polish PhD curriculum (duration, exams, dissertation…) and discuss differences and similarities with PhD programs in other EU Countries, US, and UK .
- There are important issues and relevant literature that were non considered in the Discussion. For instance, academic stress has been linked to abnormal brain plasticity in graduate students. The date is relevant because plasticity plays a pivotal role in the development of mental disorders. Please include and discuss the following reference:
- A) Concerto C. Academic stress disrupts cortical plasticity in graduate students. Stress, Volume 20, Issue 2, Pages 212 – 216. 2017”
- Health risk behaviors often reported in graduate students (use of cannabis, energy drinks, stimulants) have been linked to mental distress in graduate students. The author should include the following reference and a statement regarding the need to include other relevant variables in future studies:
- Al Sawah M. Perceived stress and coffee and energy drink consumption predict poor sleep quality in podiatric medical students: A cross-sectional study. Journal of the American Podiatric Medical Association, Volume 105, Issue 5, Pages 429 – 434, September 2015
- One important aspect that was not considered are the temperament and personality traits which have been linked to mental distress in students across programs.
- Conclusions: I would remove the bullet points
Author Response
Dear Reviewer,
Thank you for your time and preparation of the review of our article. The introduced changes are marked in the text. In response to specific comments:
Ad 1. The Introduction should be completely revised: they should focus on mental student’s mental health clearly stating the gap in the literature. Then they should indicate they research questions. As it is the Intro is very confusing, with repetitions. It should focus on education.
The introduction was corrected according Reviewer comments.
Ad 2. Some of the Results are in the Discussion section
The method of presenting the obtained results was reformulated (section Results).
Ad 3. The Discussion should focus on the results. The Author should describe the organization of the Polish PhD curriculum (duration, exams, dissertation…) and discuss differences and similarities with PhD programs in other EU Countries, US, and UK .
There are important issues and relevant literature that were non considered in the Discussion. For instance, academic stress has been linked to abnormal brain plasticity in graduate students. The date is relevant because plasticity plays a pivotal role in the development of mental disorders. Please include and discuss the following reference:
1) Concerto C. Academic stress disrupts cortical plasticity in graduate students. Stress, Volume 20, Issue 2, Pages 212 – 216. 2017”
Health risk behaviors often reported in graduate students (use of cannabis, energy drinks, stimulants) have been linked to mental distress in graduate students. The author should include the following reference and a statement regarding the need to include other relevant variables in future studies.
The cited literature was expanded.
Ad 4. One important aspect that was not considered are the temperament and personality traits which have been linked to mental distress in students across programs.
We are aware of the influence of personality traits and temperament on mental health, but the assessment of personality traits was not the aim of the presented study. The Reviewer's comment in the work limitations has been taken into account.
Sincerely,
MS